# Effects of Heat Treatments on Various Characteristics of Ready-to-Eat Zucchini Purees Enriched with Anise or Fennel

**DOI:** 10.3390/molecules27227964

**Published:** 2022-11-17

**Authors:** Luiza-Andreea Tănase (Butnariu), Oana-Viorela Nistor, Gabriel-Dănuț Mocanu, Doina-Georgeta Andronoiu, Adrian Cîrciumaru, Elisabeta Botez

**Affiliations:** 1Faculty of Food Science and Engineering, “Dunărea de Jos“ University of Galați, 111 Domnească Street, 800201 Galați, Romania; 2Cross-Border Faculty, “Dunărea de Jos” University of Galați, 111 Domnească Street, 800201 Galați, Romania

**Keywords:** RTE, zucchini puree, steaming, baking, anise, fennel, texture, color

## Abstract

Galactagogue herbs, also known as natural lactation adjuvants, are frequently used to stimulate breast milk production. Due to their antioxidant activity and phenolic content, anise (*Pimpinella anisum* L.) and fennel (*Foeniculum vulgare* L.) were chosen to increase the added value of zucchini (*Cucurbita pepo* L.) purees. At the same time, this work aimed to determine the influence of heat treatment on various characteristics of the final product. The phytochemical content, color parameters, and rheological and textural parameters of zucchini purees enriched with herbal aqueous extracts were determined after processing and after one week of storage (4 °C). In the case of antioxidant activity, samples registered a variation between 6.62 ± 1.71 and 38.32 ± 3.85 µM Trolox/g DW for the samples processed by steam convection. The total difference color parameter (ΔE) increased seven times after one week of storage compared to samples at T_0_. Fennel and anise aqueous extracts helped improve the rheological behavior of zucchini samples both by steam and hot air convection. This study may serve as a springboard for future investigations and clinical trials into the scientific validity and safety of ready-to-eat foods with special destinations.

## 1. Introduction

People tend to choose ready-to-eat (RTE) foods due to the lack of time, demanding careers, or convenience. RTE foods are generally semi-processed or totally processed products that can be reheated or consumed directly. They include a variety of products, such as cooked meat and poultry, cooked salads, smoked or salted seafood, purees, or fresh-cut fruits and vegetables [1]. The World Health Organization (WHO) estimates that 60% of adults and nearly one in three children (29% of boys and 27% of girls) in the WHO European region are obese (as of 2022) [2]. For obese women who want to become mothers, the chances of having obese children are much higher, especially if they develop gestational diabetes [3]. Therefore, a balanced and healthy lifestyle is necessary for the optimal development of both mother and child. The most recommended foods for women who are pregnant or breastfeeding include green vegetables, such as zucchini, courgettes, salads, spinach, and many others. Zucchini (*Curcubita pepo* L.) is a small summer marrow or green squash, which is usually available fresh, consumed raw in salads, or served processed in soups or purees [4]. Zucchini is very popular as it is low in calories (a medium-sized zucchini only has 25 calories), which can be attributed to its high-water content (about 96%). At the same time, zucchini has a high nutritional value—it is rich in potassium, folic acid, and provitamin A. Zucchini also contains a high percentage of magnesium, phosphorus, and vitamin C, which are needed to build and maintain healthy bones and brains. Regular consumption of zucchini helps treat asthma and can be used to prevent scurvy and bruising caused by vitamin C deficiency [5]. Even with a well-balanced diet, numerous women encounter lactation problems and appeal to natural adjuvants to stimulate lactation, such as galactagogue herbs. Insufficient breast milk production is frequently counted as the main reason for breastfeeding interruptions [6]. Among the most used and known galactagogues are anise (*Pimpinella anisum* L.) and fennel (*Foeniculum vulgare* L.) due to their various health benefits, but also to their affordable prices and high availability. *Foeniculum vulgare* L. is a biennial medicinal herb belonging to the *Apiaceae* (*Umbelliferae*) family. According to scientific studies, fennel essential oil has emmenagogue and galactagogue properties and, due to its antispasmodic potential, is a treatment for pediatric colic and respiratory disorders [7]. The most widespread Pimpinella species in the *Apiaceae* family is *Pimpinella anisum* L. Anise is used as a galactagogue all over the world due to its potent estrogenic activity, which is brought on by its two main oil constituents, trans-anethole (93.9%) and estragole (2.4%) [8]. When it comes to preparing vegetables, one of the most used processing methods is steaming; it is found to be superior to boiling and it is also known as the most popular solution for preserving nutrients. The advantages of steaming are reflected in the vegetables’ phytochemical content and texture preservation [9]. On the contrary, conventional baking methods generally include charcoal baking and oven baking. However, these methods have different disadvantages, including long processing times, food surface overheating, and nutrient losses [10]. Therefore, as the second method of processing, hot air convection was chosen. This study produced and characterized ready-to-eat products fortified with aqueous extracts from aromatic galactagogue plants, such as fennel and anise, since they are rich in phenolics and antioxidants, and improve the nutritional and sensorial properties of zucchini purees.

## 2. Results and Discussion

### 2.1. Heat-Induced Changes 

#### 2.1.1. Cooking Loss Measurements and Calculations

The results of cooking loss and cooking yield of raw zucchini after thermal treatments are presented in Table 1. 

Cooking food improves digestion and increases the absorption of nutrients, even so, the choice of the proper processing method is essential.

Cooking loss and yield determinations aim to identify the most suitable and efficient methods to cook zucchini and facilitate the purees.

The data indicated that the lowest cooking loss was obtained for the steamed sample (0.57 ± 0.09%). In direct correlation, the highest cooking yield value was also obtained for the steamed sample (99.43 ± 0.09%). These results can be attributed to the high moisture content of zucchini and the steam processing capacity to retain the nutrients in the samples, according to [11]. As expected, the hot air convection registered a cooking yield of 65.64 ± 0.46%—almost 35% lower than the one obtained via steaming.

#### 2.1.2. Differential Scanning Calorimetry

Figure 1 presents the heat flow variation of the raw zucchini exposed to a heat cycle between −20 and 150 °C obtained by using differential scanning calorimetry. 

In the DSC analysis of zucchini, the temperature ranged from −20 to 150 °C, which represents one complete cycle of zucchini exposure to heat. The DSC analysis aimed to observe the main types of denaturation that can occur during the thermal treatment. It could be observed that all peaks were endothermic. The first peak is associated with an area between 33.33 and 47 °C, which could be the reaction of some specific compound degradations or microstructure reorganization.

Zucchini, as with other fruits and vegetables, contains a mixture of different compounds, such as proteins, lipids, carbohydrates, fibers, minerals, and vitamins, which in the presence of water excess could undergo different order–disorder phase transitions [12].

From 51.33 to 74.66 °C, a slight heat flow decreases from −21.21 to −20.75 mW could be observed, which could be associated with the gelatinization peak temperatures, according to [13], mainly generated by the amount of dietary fiber of zucchini puree and by the shredding process, which contributes to granule architecture.

The melting area is clearly described by a significant peak, which begins at 82 °C and ends at 106.5 °C, contrary to the findings of [14], where the authors investigated the thermal properties of quinoa puree and did not find this type of thermal characteristic. After this, a decrease in heat flow from −78.00 to −70.41 mW was registered, which could be attributed to the glass transition status, which facilitates the transformation of amorphous food materials from their glassy state into a rubbery state, related to the safe storage and stability of the products [15].

### 2.2. Bioactive Compounds

Herbal products, due to their phytochemical content, are increasingly being used worldwide, such as in nutraceuticals and phytopharmaceuticals [16]. Table 2 presents the global phytochemical characterizations of different ready-to-eat zucchini purees enriched with herbal aqueous extracts.

As observed from Table 2, hot air convection had a lower impact on the phytochemical content of all samples, compared to water vapor convection. Therefore, the highest values for antioxidant activity, total phenolic and flavonoid content, beta-carotene, lycopene, and total carotenoids were registered by a baked sample with anise aqueous extract, EAZC, respectively (*p* < 0.05).

The addition of herbal aqueous extract was enhanced by 5 times the antioxidant activity of the samples treated by water vapor convection and about 1.5 times for those by hot air convection. Paciulli et al. [17] reported a value of 6.29 ± 1.65 μM Trolox/g DW for the antioxidant activity of the blanched zucchini puree, which is lower than the values obtained for our control samples. Therefore, it can be stated that both thermal processing methods chosen for the study are less harmful compared to the classic ones.

As mentioned earlier, the beta-carotene content registered higher values in the zucchini puree samples treated by hot air convection. According to [18,19], the degradation of beta-carotene and lycopene during processing is mainly due to isomerization and oxidation, which may depend on several factors, such as processing time, temperature, moisture content, oxygen, and light density.

After one week of storage at 4 °C, the samples registered higher or similar values to the fresh samples for all of the phytochemical contents determined. The increases in the content of flavonoids and polyphenols and, therefore, in the antioxidant activity of zucchini purees may be due to the possible hydrolysis of conjugated polyphenols, as stated by [20]. Moreover, the chemical conformation changes in the phenolic compounds during thermal treatment helped them solubilize and extract more in water during the Folin–Ciocâlteu assay, causing an increase in the total polyphenol content [21]. In addition, it was reported that thermic processing could result in cleavage of the phenolic–sugar glycosidic bonds, producing phenolic aglycons and improving reactivity with the Folin–Ciocâlteu reagent, causing an increase in the TPC and antioxidant activity, respectively [20]. Contrarily, [22] reported halving the total polyphenol content after 20 days of storage at 4 °C in the cases of dairy desserts, with encapsulated cornelian cherry, chokeberry, and blackberry juices. At the same time, the chlorophyll and total carotenoid content registered only slight decreases, proving to have good stability after one week of storage (4 °C).

### 2.3. Determination of the In Vitro Release of Phenols from Zucchini Purees 

The gastrointestinal behaviors of polyphenols under simulated digestion for 4 h are shown in Figure 2.

The negative downward slope generated by the variation of total polyphenol content in the SGJ can be traced to five out of six samples obtained from zucchini. Even so, a significant release of phenols from the matrix was observed in the gastric phase for ZM_1_, with a maximum of 3.35 ± 0.15% after 2 h of digestion.

From Figure 2b, it can be observed that ZM_1_ showed, again, the highest stability in the simulated intestinal phase, with a maximum phenol release of 9.20 ± 1.19% after 120 min of digestion, among all of the samples. Apart from this, all of the samples registered positive values, between 0.63 ±1.57 and 9.20 ± 1.19% at the end of the in vitro digestion. Previously, [23] reported higher values (up to 29.89 ± 0.26%) for the remaining phenolic content after 2 h of intestinal in vitro digestion, in different meatballs enriched with thyme or lemon balm aqueous extracts.

### 2.4. Influence of Heat Treatments on Color Evaluation of Zucchini Purees 

Color is the most pleasant characteristic of fruits and vegetables. Changes in the instrumental color parameters of zucchini puree, raw and enriched with anise or fennel aqueous extract after processing and storage, are displayed in Table 3. The average values of L* (brightness), a* (red–green component), and b* (blue–yellow component) in raw zucchini puree were 59.08 ± 0.21, −6.11 ± 0.04, and 30.92 ± 0.32. The saturation index (C^*^) was 31.51 ± 0.43 and the hue angle (H°) was 101.18 ± 0.7°. The brightness (L^*^) of zucchini puree enriched with anise or fennel aqueous extract slightly decreased within the heat treatment compared with the raw sample. The highest loss of lightness was noticed for sample EFZA (11.83%). After 7 days of storage, the L^*^ values of samples enriched with anise or fennel aqueous extract were still high and remained stable. In [24], the authors obtained similar results in the case of courgettes cooked via different gastronomy techniques. After heat processing, the steamed samples (EFZA and EAZA) became less greenish (higher—a^*^), while the samples treated by hot air convection (EFZC and EAZC) were greener (lower—a^*^) compared to the raw zucchini puree. It seems that the measure to which the green component of the color of zucchini puree changes depends on the type and time of heat processing and some compounds of fennel or anise aqueous extracts. The loss of green color is related to chlorophyll, which is a degradation and transformation in pheophytin a. This aspect is correlated with chlorophyll loss. 

Heat treatments also induced a decrease in the yellow component (b^*^); during storage, no important changes were noticed. During storage, the very important increase of the a^*^ component to positive values (33.51–35.84) revealed the appearance of a strong red hue due to the decrease in the angle (from 101.18° to values between 33.07° and 37.04°), similar to the findings presented by [25]. An increase in greenness, for avocado puree treated via the microwave, was also reported by [26]. On the other hand, [27] reported an increase in the green color in the case of purple eggplant and zucchini after steaming and sous vide treatments.

The loss of some compounds during processing, such as lutein, carotenoids, and chlorophylls, could be responsible for decreases in b^*^ values. In [25], the authors did not note any important change in the yellow color of Galega kale cooked by boiling and vacuum cooking. In the present research, the total color difference (ΔE) values were calculated relative to raw zucchini puree. After the heat treatments, the ΔE values for all samples decreased and were less than 12, which represents the reference value according to [28]. During storage, the ΔE values increased. On the seventh day of storage, the ΔE values varied between 49.37 ± 0.38 (EAZA) and 51.75 ± 0.12 (EAZC). According to the ΔE scale table, ΔE values for zucchini puree with fennel or anise aqueous extract were modified to different colors (ΔE > 12).

The hue angle (H°) values increased for steamed samples compared to raw zucchini puree. In the case of samples treated via hot air convection, the values of the hue angle (H°) decreased, thus resulting in a change of color from green to yellow. The storage period determined important losses of the hue angle relative to the control sample. Similar findings were reported by [29] in a study on the effect of the sous vide treatment on broccoli. Chroma (C*) values slightly decreased within the heat treatment compared with the raw sample. These findings could be associated with the transformation of chlorophyll into pheophytin. After 7 days of storage, the C* values increased; perhaps some compounds of fennel or anise aqueous extract contributed to these results. 

The browning index (BI), described as the brown color purity [30], is an essential indicator of the browning reaction. From Table 3, it is observable that, for both thermal processing methods, the values of the browning index increased, perhaps due to the enzymatic or non-enzymatic reactions that took place. The BI values at 7 days of storage showed a succession of EAZC (220.59 ± 1.67) > EAZA (210 ±1.28) > EFZA (203.53 ± 0.48) > EFZC (184.12 ± 1.76). The whiteness index (WI), an indicator of enzymatic discoloration, decreased after the heat treatments for all samples with fennel or anise aqueous extract, and started to increase until the end of the storage period. In [31], the authors obtained similar results for fresh-cut green beans treated with calcium chloride during storage. Another indicator related to product degradation via light, chemical exposure, and processing is the yellowness index (YI) [32]. The increased YI values for all samples with fennel or anise aqueous extracts compared to the raw zucchini puree can be attributed to the pigment deterioration associated with Maillard reactions.

### 2.5. Bostwick Consistency 

The influences of the thermal treatment and aqueous extract addition on flow resistance were analyzed via the Bostwick consistency (Figure 3).

The type of thermal treatment significantly influenced the puree consistency. The samples obtained via steam convection flowed at almost a double distance compared to samples obtained via hot air convection. This could be explained by the intensity of the transformations occurring in the plant material during processing. Previous studies revealed that the cooking method affects the vegetable structure [27]. The addition of aqueous extract increased the flow distance by approximately 30% for the samples obtained via steam convection and 8% for the samples obtained via hot air convection, respectively. This behavior may be due to water, which reduces the solid particle concentration in the serum phase and, subsequently, reduces the consistency of the vegetable puree [33]. The values of the consistency indices are in accordance with those presented by [34] for kale puree. The same authors stated that low consistency indices indicate more consistent purees with higher resistance to the flow.

### 2.6. Rheological Analysis

The dependencies of the elastic (G′) and viscous (G″) moduli by the strain and frequencies for the fresh and one-week stored puree samples are presented in Figure 4. 

All samples presented a predominant elastic character, since G′ > G″. This behavior is specific for vegetable purees, which belong to the category of concentrated structured suspensions [35]. For the fresh purees, the lowest values for G′ and G″ were registered for the samples obtained by hot air convection with no added extract. Within the same treatment, samples with added extracts presented higher values for the two moduli, while the type of extract did not influence these values significantly. After one week of storage in refrigeration conditions, the values for G′ and G″ presented slightly lower values, but the tendency remained the same. The rheological properties of vegetable purees are influenced by many factors, such as volume concentration, particle size, and chemical composition of the solid phase, as well as the viscosity, chemical composition, and electrolyte concentration of the liquid phase [13,33,35,36]. 

Vegetable purees can exhibit very complex rheological behaviors due to their structural complexities [36]. In time, they could present pseudoplastic behaviors, while in certain conditions, their rheological properties are time-dependent (thixotropic).

Figure 5 presents the time-dependent rheological behavior of zucchini purees supplemented with 6% aqueous anise/fennel extract fresh samples (a) and one-week stored samples (b). As in the case of the strain and frequency sweep test, the samples show very similar behavior, with small differences induced by the manufacturing process and the addition of aqueous extracts. The recovery degree for fresh samples ranged between 93.5% and 97.87%, while for the stored samples, this degree decreased to 80.99–84.45%, perhaps due to the structural rearrangements during storage.

### 2.7. Texture Analysis

Table 4 presents the values of the textural parameters of the pureed samples. Firmness presented higher values for the samples with no added extract (0.75 ± 0.03 N for the steam-processed sample and 0.72 ± 0.01 N for the hot air-processed sample in fresh form). The addition of aqueous extracts led to a slight reduction in firmness, perhaps due to the lower concentration of the solid phase. The opposite effect of aqueous extracts could be observed for adhesiveness, which showed approximately a 1.5 higher value when compared to the control samples. Cohesiveness and springiness did not show significant differences. Similar values were reported by the authors of [37] for carrot puree and by the authors of [38] for courgette puree. After one week of storage, the instrumental textural parameters did not significantly change, showing good stability.

### 2.8. FT-IR Analysis

In an attempt to understand the changes induced by the thermal treatments and the aqueous extract addition in the zucchini puree, an FT-IR analysis (Figure 6) was used for the investigation.

Absorption peaks in the spectra of the zucchini samples processed via hot air or steam convection shifted toward 3311.77–3319.79 cm^−1^, corresponding to O-H bound [39], which could be attributed to water content. The peak in the range of 1635.80–1635.94 cm^−1^ could be the stretching vibration of the carbonyl group (C=O) [40]. As previously mentioned by [23], the peaks corresponding to 1440 and 1635–1638 cm^−1^, which are related to trans-anethole, respectively, estragole was identified in the zucchini samples. 

Peaks at 1418.53–1558.98 cm^–1^ correspond to asymmetric and symmetric carboxylate ion stretching. The band at 1061 cm^−1^ is attributed to COC stretching vibrations [41]. No different peaks or bands were determined to compare the possible changes induced at the compositional level via the hot air convection or steam convection. Thus, it could be formulated that from the point of view of the FT-IR analysis, the processing type does not induce observable changes at the compositional level. 

### 2.9. Sensory Analysis

Figure 7 presents both the overall acceptance (a) and the diagram representing the sensory attributes specific to zucchini purees (b).

While the sensorial aspects are decisive in food acceptance, the samples were examined by the panelists in order to identify the main changes induced by the thermal treatment and the plant aqueous extract addition. 

As expected, the scores obtained for the zucchini control sample treated via hot air convection (7.1 ± 0.42) were higher than those processed by steam (6.4 ± 0.52); thus, the ZM_2_ sample was more appreciated than ZM_1_. A similar trend is followed by the samples with aqueous extract addition. 

The EFZC sample registered the highest score 7.4 ± 0.7 for the overall acceptance, on the other hand between the EAZC (7.1 ± 0.99) and ZM_2_ (7.1 ± 0.43), there were no significant differences that could be justified by the hot air convection treatment, and not depending on the influence of the extract addition. The samples EAZA and EFZA registered slight differences (of 0.6 points), which could possibly be attributed to the fennel aqueous extract.

Figure 7b presents the main sensorial attributes of the zucchini purees from the exterior aspects to specific attributes, which could be correlated with the textural analysis.

As can be observed, the exterior aspect and the colors of the zucchini purees did not exhibit noticeable differences. Taste, aroma, and mouthcoating gained similar scores for each sample. Some differences were reported between these characteristics for the EAZA sample; in particular, the aroma registered only 5.3 points corresponding to the lowest value. It could be observed that a 6% plant aqueous extract did not significantly influence the taste and aroma of zucchini purees. Similar results were reported for zucchini purees with different hydrocolloid additions [42].

## 3. Materials and Methods

### 3.1. Reagents and Chemicals

The following chemicals were used to perform the determinations for the characterization of zucchini purees: 2,2-diphenyl-1- picrylhydrazyl (DPPH), 6-hydroxy-2,5,7,8-tetramethylchromane-2-carboxylic acid (Trolox), potassium persulfate (K_2_O_2_S_8_), Folin–Ciocâlteu reagent, gallic acid, sodium carbonate (Na_2_CO_3_) 20%, quercetin, sodium nitrite (NaNO_2_) 5%, aluminum chloride (AlCl_3_) 10%, sodium hydroxide (NaOH) 1 M, methanol (HPLC grade), hexane (HPLC grade), and acetone (HPLC grade), which were all purchased from Sigma-Aldrich Steinheim, Germany.

### 3.2. Samples Preparation

#### 3.2.1. Preparation of the Aqueous Extracts of Galactagogue Herbs

Aqueous extracts of anise or fennel were obtained by the method described earlier [23].

#### 3.2.2. Preparation of Zucchini Puree Enriched with Anise or Fennel Aqueous Extract

Zucchinis sorted by diameter values (ø 3.5–4 cm) were purchased from a local supermarket (Galați, Romania) and used to obtain ready-to-eat puree. They were peeled, washed, and chopped into 2 cm-high rings, which were cut into 4 equal pieces. Zucchinis were subjected to two types of heat treatments: hot air convection at 180 °C for 35 min by using an electric oven (Indesit FIMB-51K.A-IX-PL, Lodz, Poland) and water vapor convection at 94 °C for 12 min performed via a steam cooker (Zelmer 37Z010, Warsaw, Poland). Both processing times were dependent on the core temperature of the zucchini—88 °C for the hot air convection and 92 °C for steaming, respectively. 

The processed zucchini was blended for 2 min at 1900 rpm by using a vertical mixer (Bosch ErgoMixx, Gerlingen, Germany) and mixed with an aqueous extract of anise or fennel (6%) and salt (0.5%).

#### 3.2.3. Codification 

ZM_1_ and ZM_2_ are zucchini purees treated via steam convection and hot air convection, respectively.

EFZA and EAZA are zucchini purees mixed with fennel or anise aqueous extract, processed via steam convection.

EFZC and EAZC are zucchini purees mixed with fennel or anise aqueous extract, treated via hot air convection.

### 3.3. Heat-Induced Changes 

#### 3.3.1. Cooking Loss Measurements and Calculations

For each sample, the mass was measured before and after the heat treatment. Cooking loss and cooking yield were calculated according to [43], using the equations below:(1)$$\mathrm{Cooking}\;\mathrm{loss}\;\left(\%\right)=\frac{\mathrm{Uncooked}\;\mathrm{sample}\;\mathrm{weight}-\mathrm{Cooked}\;\mathrm{sample}\;\mathrm{weight}}{\mathrm{Uncooked}\;\mathrm{sample}\;\mathrm{weight}\;}\times 100$$
(2)Cooking yield =×100

#### 3.3.2. Differential Scanning Calorimetry

The thermic measurements were performed by using a differential scanning calorimeter (DSC) for the specific heat analysis; the acquisition and evaluation of data were determined with STAR^e^ software. The samples used in the DSC analyses were placed in 160 μL aluminum crucibles with pins and lids. They were subjected to one heating–cooling cycle, in the temperature interval of −20 °C to 150 °C. The cycle was divided into seven segments. Five segments had cooling rates of 10 K/min, whereas the heating rate was 6 K/min between −20 °C and 50 °C, 2 K/min between 50 °C and 80 °C, 1 K/min between 80 °C and 150 °C, 2 isothermal processes of 2 min durations (each at −20 °C), and 150 °C temperature, respectively.

### 3.4. Global Phytochemical Characterization

The antioxidant activity, by a DPPH-free radical scavenging assay, total phenolic content (TPC) by the Folin–Ciocâlteu method, and total flavonoid content (TFC) were determined as described earlier in [23].

For the determination of beta-carotene, lycopene, chlorophyll, and total carotenoids, an extraction with n-hexane: acetone (ratio 3:1 *v*/*v*) was used. Briefly, an amount of 1 g of puree was dissolved in 10 mL of a mixture of n-hexane: acetone, in a falcon tube. The extraction was performed at a constant frequency of 40 kHz and power of 100 W. Ice was added to maintain a constant temperature at 30–40 °C in the ultrasonic bath. The samples were centrifuged at 9000× *g*, 4 °C for 5 min and spectrophotometrically read at 450 nm (total carotenoids), 470 nm (β-carotene), 503 nm (lycopene), 645 nm (chlorophyll a), and 663 nm (chlorophyll b). The contents of beta-carotene, lycopene, chlorophyll, and total carotenoids were calculated according to the following Equation (3) [44] and by using the Arnon (1949) Equation (4):(3)$$\mathrm{BC}/\mathrm{LYC}/\mathrm{TC}\;(\mathrm{mg}/g)=\frac{A\times V\times 10^{6}}{\varepsilon \times m\times 100}\times f_{d}$$
(4)$$\mathrm{Chl}\;(\mathrm{mg}/g)=0.0202\times +0.00802\times A_{663}$$
where *A* is absorbance; *V* is the volume of the final analyzed extract, mL; *ε* is the extinction coefficient (ε = 2500 for β-carotene, *ε* = 3450 for lycopene, *ε* = 2590 for total carotenoids); *m* represents the weight of the analyzed sample, g; and *f_d_* is the dilution factor. The experiments were performed in triplicate.

### 3.5. Determination of the In Vitro Release of Phenols from Zucchini Purees

In vitro digestion was performed by simulating gastric and intestinal phases [45]. The gastric phase was simulated using gastric juice (SGJ) with porcine pepsin (40 mg/mL in 0.1 M HCl, pH = 3.0). The intestinal phase was performed using simulated intestinal fluid (SIF) containing pancreatin from the porcine pancreas (2 mg/mL in 0.9 M sodium bicarbonate, pH = 7). During the experiment, the samples were incubated in a shaker (Medline Scientific, Chalgrove, Oxon, UK), at 150 rpm and 37 °C, for 2 h/ digestion phases. Every 30 min, a 0.2 mL post-hydrolysis fraction was gathered for TPC estimation to finally assess the percentage of phenols released. All the determinations were performed in triplicate.

### 3.6. Color Evaluation of Zucchini Purees

The color parameters of the purees were determined using a Minolta Chroma Meter CR-410 (Konica Minolta, Osaka, Japan). For the color analysis, each sample was stored in a Petri dish. The color parameters read were L* (lightness/darkness), a* (red/green), and b* (yellow/blue). The total color difference (ΔE), browning index (BI), yellowness index (YI), and whiteness (WI) index were determined as presented earlier in [23]. All the measurements were performed in triplicate, both after being processed and after one week of storage under refrigeration conditions (4 °C).

### 3.7. Flow Behavior 

The flow behavior was determined by two types of determinations, namely Bostwick consistency and rheology.

#### 3.7.1. Bostwick Consistency

The flow behavior of the zucchini purees was measured using the method described by [34], using a Bostwick consistometer (CR Instruments Ltd., Christchurch, UK). The puree was placed in the consistometer tank and left for equilibration for 1 min before being allowed to flow under its weight along a flat surface for 30 s at room temperature (21 ± 2 °C). The Bostwick consistency index is the distance traveled by the puree, expressed in centimeters. Each measurement was performed in duplicate.

#### 3.7.2. Rheological Analysis

The rheological behavior was studied using an AR2000ex rheometer (TA Instruments, New Castle, DE, USA). All rheological tests were conducted using plate–plate geometry, 40 mm in diameter, and a 2 mm gap. 

First, a stress sweep test at a frequency of 1 Hz was applied in order to establish the linear viscoelastic region. Then a frequency sweep test, in the range of 0.1–100 Hz, was applied to study the frequency dependencies of the elastic and viscous moduli. 

In order to study the capacities of the zucchini puree samples to recover the deformations occurring during production and packaging processes, a three-interval thixotropy test (3ITT) was applied. In the first interval of 5 min, a low strain (approximative 0.1%) was applied at a frequency of 1 Hz. In the second 5 min interval, 100% strain was applied at a frequency of 1 Hz. The third interval lasted 10 min; the strain returned to approximately 0.1%. The recovery degree of the samples was calculated using Equation (5):(5)$$\%Rec=\frac{{G^{\prime }}_{10}}{{G^{\prime }}_{1}}\times 100$$
where *G′*_1_ represents the initial value of the elastic modulus, while *G′*_10_ represents the value of the elastic modulus after 10 min from the beginning of the test.

All measurements were made in duplicate under a controlled temperature (25 °C).

### 3.8. Texture Analysis

The instrumental analysis of the texture was performed using the tpa (texture profile analysis) method, which consists of double penetration. The Brookfield CT3 1000 texture analyzer was used for testing. The purees were packed in plastic containers, 35 mm in diameter and 50 mm in height (40 mm height samples). A polyacrylic cylinder with a diameter of 24 mm was used for double penetration of the samples, up to a depth of 10 mm, with a speed of 1 mm/s. For data recording and processing, the texture analyzer was connected to a computer using the TexturePro CT V1.5 program in order to determine the textural parameters: firmness, adhesiveness, cohesiveness, and springiness. Each determination was performed in triplicate.

### 3.9. FT-IR Analysis

The infrared spectra were collected using a Nicolet iS50 FT-IR spectrometer (Thermo Scientific, Waltham, MA, USA) equipped with a built-in ATR accessory, DTGS detector, and KBr beam splitter. Therefore, 32 scans were co-added over the range of 4000–400 cm^−1^ with a resolution of 4 cm^−1^. Air was used as a reference for the background spectrum before every sample. After each spectrum, the ATR plate was cleaned using an ethanol solution. To validate that no residue from the previous sample remained, a background spectrum was collected each time and compared to the last spectrum background. An FT-IR spectrometer was held in a room under a controlled temperature (21 °C).

### 3.10. Sensory Analysis 

Sensory analysis was performed by a group of 10 untrained panelists from the Food Science and Engineering staff (2 males and 8 females), at room temperature (21 ± 2 °C). The panelists were non-smokers between the ages of 25 and 55 years old. The sample preparation and processing method, alongside the benefits for the human body, were clearly explained before participation. Samples were served in random order to each panelist using transparent plastic containers, coded with random numbers. The intensity of each attribute was determined using a nine-point scale, where 1 was the weakest/most unpleasant perception and 9 was the strongest/most pleasant perception [46]. The tasters noted the following sensory attributes: exterior acceptance, color, taste, aroma, aftertaste, mouthcoating, firmness, consistency, cohesiveness, and overall acceptance. 

### 3.11. Statistical Analysis

All analyses were conducted in at least two replicates and data are reported as mean ± standard deviation (SD). To identify significant differences, all experimental data were subjected to one-way analysis of variance (ANOVA), using Minitab 17 statistical software. Tukey test was performed to identify significant differences between the results (*p* < 0.05).

## 4. Conclusions

This study shows that thermal treatments and the addition of herbal aqueous extracts could offer new possibilities for food acceptance. The results indicate that aqueous extracts have statistically significant impacts (*p* < 0.05) on the phytochemical content of zucchini purees. According to the research, herbal aqueous extracts can potentially be used for the manufacturing of functional foods that primarily focus on lactogenic activity as measured by FT-IR analysis. 

The heat flow of food depends on the composition, structure, and temperature (according to our study and related to the scientific literature).

Further studies could establish if these effects are longer-lasting; zucchini purees could be included in the diets of other individuals with special needs. 

## Figures and Tables

**Figure 1 molecules-27-07964-f001:**
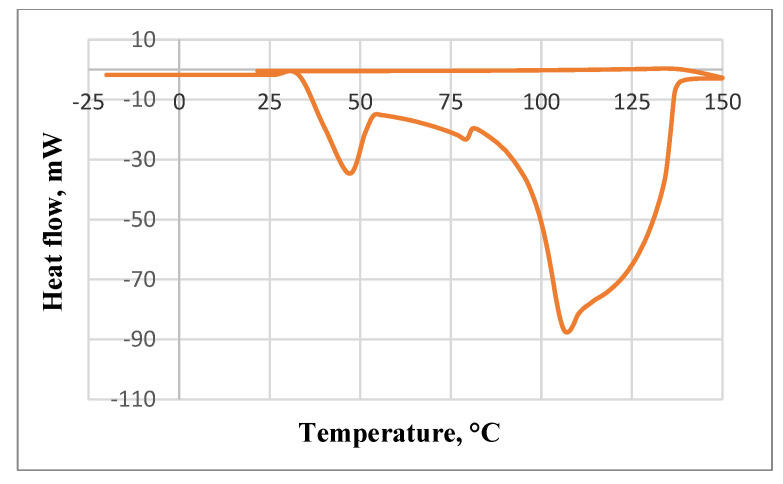
DSC thermogram (variation of heat flow as a function of temperature) of raw zucchini.

**Figure 2 molecules-27-07964-f002:**
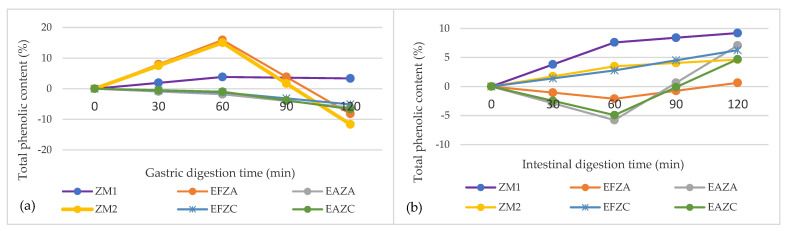
In vitro digestion of the remaining phenolic content in zucchini purees enriched with anise or fennel aqueous extract, in simulated gastric juice (**a**), and simulated intestinal juice (**b**). ZM_1_ and ZM_2_ are zucchini purees treated by steam convection and hot air convection, respectively; EFZA and EAZA are zucchini purees mixed with fennel or anise aqueous extract, processed by steam convection; EFZC and EAZC are zucchini purees mixed with fennel or anise aqueous extract, treated by hot air convection.

**Figure 3 molecules-27-07964-f003:**
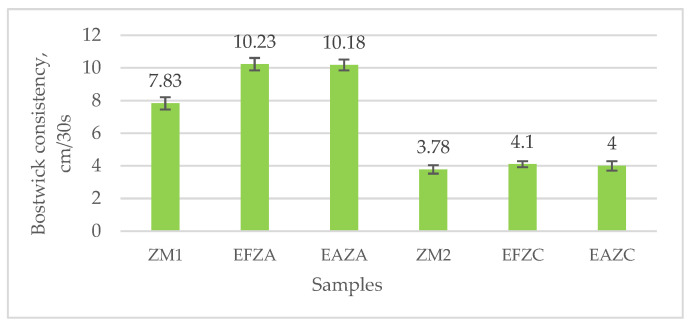
Bostwick consistency (cm/30 s) of zucchini purees enriched with anise/fennel aqueous extract, processed by steam and hot air convection. ZM_1_ and ZM_2_ are zucchini purees treated by steam convection and hot air convection, respectively; EFZA and EAZA are zucchini purees mixed with fennel or anise aqueous extract, processed by steam convection; EFZC and EAZC are zucchini purees mixed with fennel or anise aqueous extract, treated by hot air convection.

**Figure 4 molecules-27-07964-f004:**
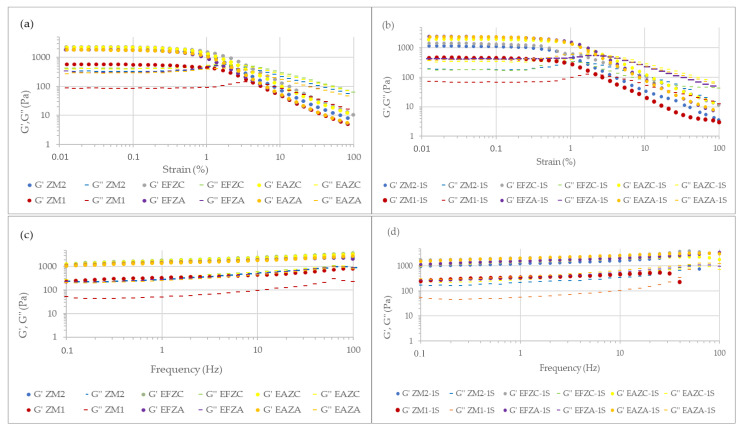
Strain (**a**,**b**) and frequency (**c**,**d**) dependent viscoelastic characteristics of zucchini purees fresh samples (**a**,**c**) and one-week stored samples (**b**,**d**). ZM_1_ and ZM_2_ are zucchini purees treated by steam convection and hot air convection, respectively; EFZA and EAZA are zucchini purees mixed with fennel or anise aqueous extract, processed by steam convection; EFZC and EAZC are zucchini purees mixed with fennel or anise aqueous extract, treated by hot air convection; “-1S” is added at each sample to differentiate one week of storage at refrigeration temperature.

**Figure 5 molecules-27-07964-f005:**
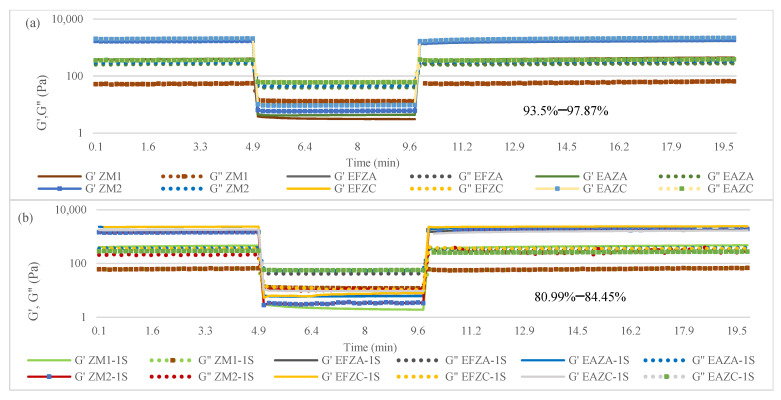
Time-dependent rheological behavior of zucchini purees supplemented with 6% aqueous anise/fennel extract fresh samples (**a**) and one-week stored samples (**b**). ZM_1_ and ZM_2_ are zucchini purees treated by steam convection and hot air convection, respectively; EFZA and EAZA are zucchini purees mixed with fennel or anise aqueous extract, processed by steam convection; EFZC and EAZC are zucchini purees mixed with fennel or anise aqueous extract, treated by hot air convection; “-1S” is added at each sample to differentiate one week of storage at refrigeration temperature.

**Figure 6 molecules-27-07964-f006:**
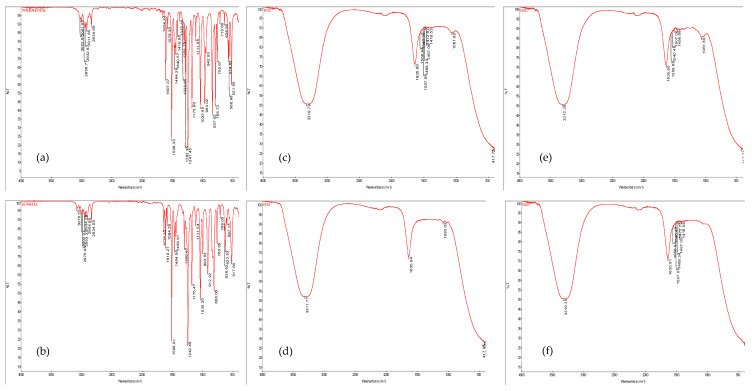
Fourier-transform infrared spectroscopy spectra of (**a**) standard trans-anethole (**b**) standard estragole (**c**) EFZC—zucchini puree mixed with fennel aqueous extract, processed by hot air convection; (**d**) EFZA—zucchini puree mixed with fennel aqueous extract, processed by steam convection; (**e**) EAZC—zucchini puree mixed with anise aqueous extract, processed by hot air convection; (**f**) EAZA—zucchini puree mixed with anise aqueous extract, processed by steam convection.

**Figure 7 molecules-27-07964-f007:**
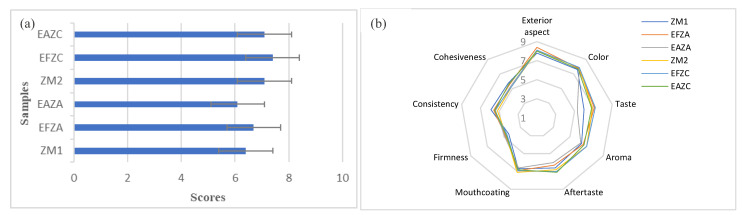
Overall acceptance (**a**) and comparative diagram of the sensory attributes specific to purees (**b**). ZM_1_ and ZM_2_ are zucchini purees treated by steam convection and hot air convection, respectively; EFZA and EAZA are zucchini purees mixed with fennel or anise aqueous extract, processed by steam convection; EFZC and EAZC are zucchini purees mixed with fennel or anise aqueous extract, treated by hot air convection.

**Table 1 molecules-27-07964-t001:** Influence of hot air and steam convection on raw zucchini.

Parameters	Hot Air Convection	Steam Convection
**Cooking loss, %**	34.36 ± 0.46 ^A^	0.57 ± 0.09 ^B^
**Cooking yield, %**	65.64 ± 0.46 ^B^	99.43 ± 0.09 ^A^

The averages on the same lines with different superscripts (A,B) are statistically significantly different (*p* < 0.05).

**Table 2 molecules-27-07964-t002:** Phytochemical content of zucchini purees enriched with fennel or anise aqueous extract, processed by steam or hot air convection, just after treatment and after one week of storage.

	T_0_					
	*Steam Convection*			*Hot Air Convection*		
	ZM_1_	EFZA	EAZA	ZM_2_	EFZC	EAZC
Antioxidant Activity, µM Trolox/g DW	6.62 ± 1.71 ^D^	29.64 ± 1.43 ^C^	38.32 ± 3.85 ^B^	33.76 ± 1.92 ^B, C^	49.55 ± 0.34 ^A^	55.1 ± 3.75 ^A^
TPC, mg GAE/g DW	17.27 ± 0.11 ^E^	17.96 ± 0.11 ^D, E^	20.1 ± 0.17 ^C^	18.3 ± 0.45 ^D^	22.11 ± 0.39 ^B^	25.68 ± 0.63 ^A^
TFC, mg EQ/g DW	15.89 ± 0.99 ^B, C^	14.76 ± 1.21 ^B, C^	14.3 ± 0.66 ^C^	17.71 ± 0.59 ^B^	21.43 ± 1.39 ^A^	24.37 ± 1.68 ^A^
BC, mg/g DW	2.97 ± 0.27 ^C^	3.07 ± 0.24 ^C^	2.94 ± 0.30 ^C^	4.64 ± 0.07 ^B^	4.91 ± 0.0 ^B^	5.93 ± 0.26 ^A^
LYC, mg/g DW	1.65 ± 0.13 ^D^	1.76 ± 0.02 ^C,D^	1.67 ± 0.18 ^D^	2.09 ± 0.05 ^B,C^	2.17 ± 0.02 ^B^	2.69 ± 0.19 ^A^
TC, mg/g DW	3.11 ± 0.24 ^C^	3.29 ± 0.27 ^C^	3.08 ± 0.33 ^C^	5.72 ± 0.06 ^B^	5.33 ± 0.33 ^B^	7.57 ± 0.08 ^A^
Chl, µg/g DW	1.41 ± 0.00 ^B^	1.51 ± 00 ^A, B^	1.41 ± 0.00 ^B^	1.73 ± 00 ^A^	1.44 ± 0.00 ^B^	1.51 ± 0.00 ^A, B^
	**T_7_**					
	** *Steam Convection* **			** *Hot Air Convection* **		
Antioxidant Activity, µM Trolox/g DW	33.8 ± 0.75 ^C^	65.38 ± 1.12 ^B^	65.54 ± 0.00 ^B^	11.49 ± 4.88 ^C^	69.85 ± 24.04 ^A, B^	96.44 ± 6.42 ^A^
TPC, mg GAE/g DW	15.19 ± 0.2 ^E^	22.33 ± 0.09 ^C^	18.11 ± 0.13 ^D^	26.8 ± 0.11 ^B^	28.5 ± 1.55 ^B^	34.09 ± 0.45 ^A^
TFC, mg EQ/g DW	17.5 ± 1.38 ^D, E^	20.35 ± 1.3 ^C, D^	14.67 ± 0.22 ^E^	23.48 ± 1.18 ^B, C^	24.23 ± 2.6 ^B^	31 ± 0.4 ^A^
BC, mg/g DW	2.67 ± 0.00 ^E^	3.09 ± 0.05 ^D^	2.44 ± 0.11 ^F^	5.7 ± 0.1 ^B^	4.12 ± 0.1 ^C^	7.63 ± 0.05 ^A^
LYC, mg/g DW	1.42 ± 0.02 ^D^	2.15 ± 0.06 ^C^	1.72 ± 0.07 ^D^	2.96 ± 0.07 ^B^	2.65 ± 0.11 ^B^	4.61 ± 0.3 ^A^
TC, mg/g DW	2.8 ± 0.02 ^E^	3.09 ± 0.00 ^D^	2.35 ± 0.11 ^F^	5.02 ± 0.07 ^B^	4.1 ± 0.09 ^C^	6.4 ± 0.29 ^A^
Chl, µg/g DW	1.37 ± 0.00 ^B^	1.37 ± 0.00 ^B^	1.33 ± 0.00 ^B^	1.34 ± 0.00 ^B^	1.34 ± 0.00 ^B^	1.50 ± 0.00 ^A^

The averages on the same line with different superscripts (A–F) are statistically significantly different (*p* < 0.05). ZM_1_ and ZM_2_ are zucchini purees treated by steam convection and hot air convection, respectively; EFZA and EAZA are zucchini purees mixed with fennel or anise aqueous extract, processed by steam convection; EFZC and EAZC are zucchini purees mixed with fennel or anise aqueous extract, treated by hot air convection; TPC—Total Phenolic Content; TFC—Total Flavonoid Content; BC—beta-carotene; LYC—lycopene; TC—Total Carotenoids; Chl—total chlorophyll; T_0_—fresh samples; T_7_—samples after one week of storage at 4 °C.

**Table 3 molecules-27-07964-t003:** Influence of heat treatments and storage on color parameters of zucchini purees enriched with anise or fennel aqueous extracts.

T_0_					
	Raw	EFZA	EAZA	EFZC	EAZC
L^*^	59.08 ± 0.21 ^A^	52.09 ± 0.19 ^B^	52.29 ± 0.64 ^B^	52.97 ± 0.73 ^B^	53.2 ± 0.02 ^B^
a^*^	−6.11 ± 0.04 ^C^	−7.28 ± 0.02 ^E^	−6.99 ± 0.02 ^D^	−5.13 ± 0.03 ^B^	−4.73 ± 0.01 ^A^
b^*^	30.92 ± 0.32 ^B^	32.56 ± 0.30 ^A^	29.30 ± 0.57 ^C^	28.49 ± 0.34 ^C^	30.69 ± 0.09 ^B^
ΔE	-	7.28 ± 0.11 ^A^	7.05 ± 0.74 ^A,B^	6.65 ± 0.54 ^A, B^	6.04 ± 0.02 ^B^
C^*^	31.51 ± 0.43 ^A^	31.36 ± 0.28 ^A^	30.12 ± 0.55 ^C^	28.95 ± 0.34 ^D^	31.05 ± 0.09 ^B^
H°	101.18 ± 0.78 ^B^	102.61 ± 0.25 ^B^	103.42 ± 0.49 ^B^	100.21 ± 0.57 ^A^	98.77 ± 0.81 ^C^
BI	61.76 ± 0.87 ^A^	79.41 ± 0.82 ^A^	66.47 ± 0.89 ^C^	65.29 ± 2.44 ^C^	73.59 ± 0.32 ^B^
WI	61.65 ± 0.47 ^B^	58.38 ± 0.01 ^C^	56.43 ± 0.24 ^B^	55.23 ± 0.80 ^A^	156.17 ± 0.03 ^A, B^
YI	74.76 ± 0.59 ^A^	89.29 ± 0.49 ^A^	80.04 ± 0.57 ^B^	76.84 ± 1.96 ^C^	82.41 ± 0.21 ^B^
T_7_					
L^*^	59.08 ± 0.21 ^A^	31.96 ± 0.24 ^B^	32.37 ± 0.15 ^B^	30.35 ± 0.61 ^C^	29.42 ± 0.15 ^D^
a^*^	−6.11 ± 0.04 ^C^	34.86 ± 0.49 ^B^	35.18 ± 0.36 ^A,B^	33.51 ± 0.48 ^C^	35.84 ± 0.04 ^A^
b^*^	30.92 ± 0.32 ^B^	25.54 ± 0.21 ^B, C^	26.55 ± 0.06 ^B^	21.82 ± 0.94 ^D^	24.74 ± 0.02 ^C^
ΔE	-	49.43 ± 0.25 ^B^	49.37 ± 0.38 ^B^	49.78 ± 0.56 ^B^	51.75 ± 0.12 ^A^
C^*^	31.51 ± 0.43 ^A^	43.21 ± 0.52 ^A^	44.07 ± 0.25 ^A^	39.98 ± 0.91 ^B^	43.55 ± 0.04 ^A^
H°	101.18 ± 0.78 ^B^	36.22 ± 0.31 ^C^	37.04 ± 0.22 ^C^	33.07 ± 0.35 ^A^	34.61 ± 0.43 ^B^
BI	61.76 ± 0.87 ^A^	203.53 ± 0.48 ^B^	210 ±1.28 ^A, B^	184.12 ± 1.76 ^C^	220.59 ± 1.67 ^A^
WI	61.35 ± 0.47 ^B^	80.60 ± 0.07 ^A^	80.81 ± 0.26 ^A^	80.31 ± 0.98 ^A^	82.93 ± 0.15 ^B^
YI	74.76 ± 0.59 ^A^	114.16 ± 0.08 ^A^	117.54 ± 0.28 ^A^	102.89 ± 0.47 ^B^	120.18 ± 0.69 ^A^

The averages on the same lines with different superscripts (A, B, C, and D) are statistically significantly different (*p* < 0.05). L*—clarity/brightness; a*—red/green color component; b*—blue/yellow color component; ΔE—total color difference; C*—chroma; h*—hue angle; BI—browning index; WI—whiteness index; YI—yellowness index; EFZA and EAZA are zucchini purees mixed with fennel or anise aqueous extract, processed by steam convection; EFZC and EAZC are zucchini purees mixed with fennel or anise aqueous extract, treated by hot air convection; T_0_—fresh samples; T_7_—samples after one week of storage at 4 °C.

**Table 4 molecules-27-07964-t004:** Instrumental determination of textural parameters of zucchini purees enriched with anise or fennel aqueous extract, processed by hot air or steam convection.

	T_0_					
	ZM_1_	EFZA	EAZA	ZM_2_	EFZC	EAZC
Firmness, N	0.75 ± 0.03 ^A^	0.5 ± 0.02 ^C^	0.51 ± 0.02 ^C^	0.72 ± 0.01 ^A^	0.52 ± 0.03 ^C^	0.60 ± 0.03 ^B^
Adhesiveness, mJ	1.19 ± 0.31 ^A, B^	1.36 ± 0.02 ^A, B^	1.87 ± 0.61 ^A, B^	1.05 ± 0.11 ^B^	2.21 ± 0.55 ^A^	2.22 ± 0.31 ^A^
Cohesiveness	0.66 ± 0.06 ^A^	0.64 ± 0.02 ^A^	0.57 ± 0.09 ^A^	0.66 ± 0.06 ^A^	0.68 ± 0.11 ^A^	0.65 ± 0.03 ^A^
Springiness, mm	7.65 ± 0.58 ^A^	8.66 ± 0.48 ^A^	8.17 ± 0.38 ^A^	8.49 ± 0.73 ^A^	8.32 ± 0.36 ^A^	8.35 ± 0.21 ^A^
	T_7_					
Firmness, N	0.65 ± 0.04 ^A^	0.52 ± 0.06 ^B^	0.50 ± 0.02 ^B^	0.64 ± 0.04 ^A^	0.55 ± 0.01 ^A, B^	0.59 ± 0.06 ^A, B^
Adhesiveness, mJ	1.39 ± 0.47 ^B, C^	1.81 ± 0.24 ^B, C^	1.91 ± 0.19 ^B, C^	1.29 ± 0.14 ^C^	2.17 ± 0.09 ^B^	3.52 ± 0.42 ^A^
Cohesiveness	0.64 ± 0.06 ^A, B^	0.58 ± 0.03 ^B, C^	0.49 ± 0.04 ^C^	0.76 ± 0.12 ^A^	0.66 ± 0.01 ^A, B^	0.58 ± 0.04 ^B, C^
Springiness, mm	7.37 ± 0.97 ^A^	8.26 ± 0.26 ^A^	7.4 ± 0.48 ^A^	8.43 ± 0.43 ^A^	8.62 ± 0.21 ^A^	8.44 ± 0.41 ^A^

The averages on the same lines with different superscripts (A, B, and C) are statistically significantly different (*p* < 0.05). ZM_1_ and ZM_2_ are zucchini purees treated by steam convection and hot air convection, respectively; EFZA and EAZA are zucchini purees mixed with fennel or anise aqueous extract, processed by steam convection; EFZC and EAZC are zucchini purees mixed with fennel or anise aqueous extract, treated by hot air convection.

## Data Availability

The data presented in this study are available on request from the corresponding author.

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
