# Peer review of "Effects of Heat Treatments on Various Characteristics of Ready-to-Eat Zucchini Purees Enriched with Anise or Fennel"

_molecules, 2022, doi:10.3390/molecules27227964_

Round 1

Reviewer 1 Report

Manuscript ID: molecules-1987410

Major remarks

1) The title is too general and does not provide the purpose of the article. It should be written as follows:

Effects of Heat Treatments on Various Characteristics of Ready-to-Eat Zucchini Purees Enriched with Anise or Fennel 

2) Numerical references should not be part of the sentence syntax. This is parenthetical content. Some examples:

-Instead of “[17] reported for the antioxidant activity of the blanched zucchini puree a value of 6.29 ± 1.65 μM Trolox/g DW, which is lower than the values obtained for our control samples.”, write “Paciulli et al. [17] reported for the antioxidant activity of the blanched zucchini puree a value of 6.29 ± 1.65 μM Trolox/g DW, which is lower than the values obtained for our control samples.” or For the antioxidant activity of the blanched zucchini puree, a value of 6.29 ± 1.65 μM Trolox/g DW was reported [17], which is lower than the values obtained for our control samples.”.

-Instead of “Similar results were reported by [42] for zucchini puree with different hydrocolloids addition.”, write “Similar results were reported for zucchini puree with different hydrocolloids addition [42].”.

-Instead of “Aqueous extracts of anise or fennel were obtained by the method described earlier in [23].”, write “Aqueous extracts of anise or fennel were obtained by the method described earlier [23]”.

- Instead of “In vitro digestion was performed as described by [45] simulating gastric and intestinal phases.”, write “In vitro digestion was performed by simulating gastric and intestinal phases [45].”.

3) Give more detailed conclusions.

Minor remarks

1)      Keywords: Instead of “RTE, characterization, zucchini, texture, aqueous extract, color parameters”, write “RTE, zucchini puree, steaming, baking, anise, fennel, texture, color”.

2)      Line 13: Instead of “Curcubita”, write “Cucurbita”.

3)      Table 2: Why does the antioxidant activity in sample ZM2 (Hot air convection) decrease after 7 days, while in all other samples it increases?

4)      Line 141: Instead of: “Phytochemical content of zucchini purees enriched with fennel or anise aqueous extract, processed by steam or hot air convection”, write “Phytochemical content of zucchini purees enriched with fennel or anise aqueous extract, processed by steam or hot air convection, just after treatment and after one week of storage”.

5)      Line 153: Instead of “enhanced the by 5 times”, write enhanced by 5 times”.

6)      Line 212: Instead of “(EFZC and EAZC) are more reddish (lower – a*)”, write “(EFZC and EAZC) are more greenish (lower – a*)”

7)   Lines 219-220: Instead of “Storage also determined an increase of red/green color component, which was higher in the samples with fennel or anise aqueous extract.”, write “During storage, the very important increase of the a* component to positive values (33.51-35.84) revealed the appearance of a strong red hue due to ... " (explain).

8)    Line 255: Instead of “Influence of heat treatments”, write “Influence of heat treatments and storage”.

9)      Line 259: Instead of “chrome”, write “chroma”.

10)   Line 260: Instead of: “whitening index”, write “whiteness index”.

11)   Line 481: Instead of “1900 rot/min”, write “1900 rpm”.

12)   Line 496: Remove parentheses.

13)   Line 504: Instead of “between -20 °C÷ 50 °C, 2 K/min between 50 °C ÷ 80 °C and 1 K / between 80 °C /  150 °C”, write “between –20°C and 50°C, 2 K/min between 50°C and 80°C and 1 K/min between 80°C and 150°C”.

14)   Line 517: Instead of “9000×g”, write “9000×g” (g in italics).

15)   Line 542: Instead of “browning (BI), yellowness (YI)”, write “browning index (BI), yellowness index (YI)”.

16)   Line 577: Instead of “textural analyzer”, write “texture analyzer”.

17)   Line 600: Instead of “using a hedonic nine-point scale”, write “using a nine-point scale”. The term “hedonic” is reserved only for acceptance tests, and not for quantitative descriptive ones.

Author Response

Review of MS: molecules - 1987410 entitled “Ready-To-Eat Zucchini Purees Enriched with Anise or Fennel Aqueous Extract Processed by Steam or Hot Air Convection”.

The authors would like to thank for the suggestions and the time spent to review the manuscript. We hope that the answers will clarify the scientifical aspects you pointed out in the manuscript.

The answers/changes are written with blue in the manuscript.

R1#

Major remarks

Comment: The title is too general and does not provide the purpose of the article. It should be written as follows: Effects of Heat Treatments on Various Characteristics of Ready-to-Eat Zucchini Purees Enriched with Anise or Fennel.

Answer: The title was changed as suggested: “Effects of Heat Treatments on Various Characteristics of Ready-to-Eat Zucchini Purees Enriched with Anise or Fennel “.

Comment: Numerical references should not be part of the sentence syntax. This is parenthetical content. Some examples: -Instead of “[17] reported for the antioxidant activity of the blanched zucchini puree a value of 6.29 ± 1.65 μM Trolox/g DW, which is lower than the values obtained for our control samples.”, write “Paciulli et al. [17] reported for the antioxidant activity of the blanched zucchini puree a value of 6.29 ± 1.65 μM Trolox/g DW, which is lower than the values obtained for our control samples.” or “For the antioxidant activity of the blanched zucchini puree, a value of 6.29 ± 1.65 μM Trolox/g DW was reported [17], which is lower than the values obtained for our control samples.”.

Answer: We have considered your suggestions regarding the numerical references and changed it in the manuscript.

Comment: Instead of “Similar results were reported by [42] for zucchini puree with different hydrocolloids addition.”, write “Similar results were reported for zucchini puree with different hydrocolloids addition [42].”.

Answer: We have considered your suggestion and changed it in the manuscript.

Comment: Instead of “Aqueous extracts of anise or fennel were obtained by the method described earlier in [23].”, write “Aqueous extracts of anise or fennel were obtained by the method described earlier [23]”.

Answer: We have considered your suggestion and changed it in the manuscript.

Comment: Instead of “In vitro digestion was performed as described by [45] simulating gastric and intestinal phases.”, write “In vitro digestion was performed by simulating gastric and intestinal phases [45].”.

Answer: We have considered your suggestion and changed it in the manuscript.

Comment: Give more detailed conclusions.

Answer: We have rewritten the conclusions.

“This study shows that both thermal treatment as well as the addition of herbal aqueous extracts could offer new possibilities of foods acceptance. The results indicate that aqueous extracts have a statistically significant impact (p<0.05) on phytochemical content of zucchini purees. According to the research, herbal aqueous extracts provide intriguing potential for the manufacturing of functional foods that have a particular focus on lactogenic activity as measured by FT-IR analysis.

Related to the scientific literature and according to our study the heat flow of food depends on their composition, structure, and temperature.

Further studies could establish if these effects are longer-lasting, and the zucchini purees could be included in the diet of other persons with special needs.”

Minor remarks

Comment:  Keywords: Instead of “RTE, characterization, zucchini, texture, aqueous extract, color parameters”, write “RTE, zucchini puree, steaming, baking, anise, fennel, texture, color”.

Answer: The authors have modified the keywords as you suggested.

Comment: Line 13: Instead of “Curcubita”, write “Cucurbita”.

Answer: The authors have corrected the Latin name of courgette.

Comment: Table 2: Why does the antioxidant activity in sample ZM2 (Hot air convection) decrease after 7 days, while in all other samples it increases?

Answer: According to scientific studies, loss of antioxidant activity during storage may be related to the softening of the vegetable matrix, favoring diffusion of the antioxidant compounds to the water, and to increased exposure of these compounds to oxidation caused by the residual oxygen in the packaging.

Comment:  Line 141: Instead of: “Phytochemical content of zucchini purees enriched with fennel or anise aqueous extract, processed by steam or hot air convection”, write “Phytochemical content of zucchini purees enriched with fennel or anise aqueous extract, processed by steam or hot air convection, just after treatment and after one week of storage”.

Answer: The authors have modified as you suggested.

Comment: Line 153: Instead of “enhanced the by 5 times”, write “enhanced by 5 times”.

Answer: The authors have modified as you suggested.

Comment: Line 212: Instead of “(EFZC and EAZC) are more reddish (lower – a*)”, write “(EFZC and EAZC) are more greenish (lower – a*)”.

Answer: The authors have modified as you suggested.

Comment: Lines 219-220: Instead of “Storage also determined an increase of red/green color component, which was higher in the samples with fennel or anise aqueous extract.”, write “During storage, the very important increase of the a* component to positive values (33.51-35.84) revealed the appearance of a strong red hue due to ... " (explain).

Answer: The authors have modified as you suggested.

“During storage, the very important increase of the a* component to positive values (33.51-35.84) revealed the appearance of a strong red hue due to the decrease of the angle (from 101.18° to values between 33.07 °and 37.04°), similar to the findings presented by [25]. “

Comment: Line 255: Instead of “Influence of heat treatments”, write “Influence of heat treatments and storage”.

Answer: The authors have modified the table’s name as you suggested.

Comment: Line 259: Instead of “chrome”, write “chroma”.

Answer: The authors have modified as you suggested.

Comment: Line 260: Instead of: “whitening index”, write “whiteness index”.

Answer: The authors have modified as you suggested.

Comment: Line 481: Instead of “1900 rot/min”, write “1900 rpm”.

Answer: The authors have modified as you suggested.

Comment: Line 496: Remove parentheses.

Answer: The authors have modified as you suggested.

Comment: Line 504: Instead of “between -20 °C÷ 50 °C, 2 K/min between 50 °C ÷ 80 °C and 1 K / between 80 °C /  150 °C”, write “between –20°C and 50°C, 2 K/min between 50°C and 80°C and 1 K/min between 80°C and 150°C”.

Answer: The authors have modified as you suggested.

Comment: Line 517: Instead of “9000×g”, write “9000×g” (g in italics).

Answer: The authors have modified as you suggested.

Comment: Line 542: Instead of “browning (BI), yellowness (YI)”, write “browning index (BI), yellowness index (YI)”.

Answer: The authors have modified as you suggested.

Comment: Line 577: Instead of “textural analyzer”, write “texture analyzer”.

Answer: The authors have modified as you suggested.

Comment: Line 600: Instead of “using a hedonic nine-point scale”, write “using a nine-point scale”. The term “hedonic” is reserved only for acceptance tests, and not for quantitative descriptive ones.

Answer: The authors have modified as you suggested.

Reviewer 2 Report

The study aimed to propose new food products, namely ready-to-eat food zucchini purees that were enriched with anise or fennel aqueous extract. The anise or fennel aqueous extract contribute to the increase of the nutritional and sensorial value of zucchini purees and due to their galactogogue properties could stimulate breastmilk production.

In the paper was also investigated the influence of heat treatment procedure (steam convection and hot air convection) and also of the storage on the characteristics of the proposed food product. The following properties of the zucchini purees enriched with herbal aqueous extract were determined: phytochemical content, color parameters, rheological and textural parameters. The results of the study were compared to other similar studies.

The conclusions do not fully reflect the research results of the present study. I recommend the reformulation of conclusions.

Author Response

Review of MS: molecules - 1987410 entitled “Ready-To-Eat Zucchini Purees Enriched with Anise or Fennel Aqueous Extract Processed by Steam or Hot Air Convection”.

The authors would like to thank for the suggestions and the time spent to review the manuscript.

We are glad that you have contributed to the improvement of our article.

The answers/changes are written with green in the manuscript.

R2#

Comment: The study aimed to propose new food products, namely ready-to-eat food zucchini purees that were enriched with anise or fennel aqueous extract. The anise or fennel aqueous extract contribute to the increase of the nutritional and sensorial value of zucchini purees and due to their galactogogue properties could stimulate breastmilk production.

In the paper was also investigated the influence of heat treatment procedure (steam convection and hot air convection) and also of the storage on the characteristics of the proposed food product. The following properties of the zucchini purees enriched with herbal aqueous extract were determined: phytochemical content, color parameters, rheological and textural parameters. The results of the study were compared to other similar studies.

The conclusions do not fully reflect the research results of the present study. I recommend the reformulation of conclusions.

Answer: The authors have detailed/reformulated the conclusions as you suggested.

This study shows that both thermal treatment as well as the addition of herbal aqueous extracts could offer new possibilities of foods acceptance. The results indicate that aqueous extracts have a statistically significant impact (p<0.05) on phytochemical content of zucchini purees. According to the research, herbal aqueous extracts provide intriguing potential for the manufacturing of functional foods that have a particular focus on lactogenic activity as measured by FT-IR analysis.

Related to the scientific literature and according to our study the heat flow of food depends on their composition, structure, and temperature.

Further studies could establish if these effects are longer-lasting, and the zucchini purees could be included in the diet of other persons with special needs.
